# The Genetic and Epigenetic Arms of Human Ageing and Longevity

**DOI:** 10.3390/biology14010092

**Published:** 2025-01-18

**Authors:** Elena Ciaglia, Francesco Montella, Valentina Lopardo, Cristina Basile, Roberta Maria Esposito, Clara Maglio, Roberta Longo, Anna Maciag, Annibale Alessandro Puca

**Affiliations:** 1Molecular and Clinical Pathophysiology Lab, Department of Medicine, Surgery and Dentistry “Scuola Medica Salernitana”, University of Salerno, Via Salvatore Allende, 84081 Baronissi Salerno, Italy; fmontella@unisa.it (F.M.); vlopardo@unisa.it (V.L.); cbasile@unisa.it (C.B.); robeesposito@unisa.it (R.M.E.); clmaglio@unisa.it (C.M.); rolongo@unisa.it (R.L.); 2Cardiovascular Research Unit, IRCCS MultiMedica, 20138 Milan, Italy; anna.maciag@multimedica.it

**Keywords:** prolonged lifespan, genes, epigenetic changes

## Abstract

To live a long and healthy life appears to be rooted in the genetic makeup of certain people. The results of research on the main genetic and epigenetic changes influencing Human Longevity are covered in this brief overview.

## 1. Introduction

Increased longevity is the greatest achievement of modern medicine, health systems, and education. A huge drop in infant mortality, due to the progress of the health care system, is probably the most significant factor contributing to the recent increase in lifespan. In addition, adopting healthy lifestyle approaches contributes to delaying ageing and increasing longevity, thus explaining the increased lifespan observed over the past century in industrialized countries [1]. Thus, a person’s environment and lifestyle, along with genetics, influence how they age. It is reasonable that a nutritious diet, regular exercise, and sufficient sleep could improve health, delay ageing, and increase longevity to a certain extent, and with a favourable genetic makeup, this would translate into a great improvement. This concept can be translated at the cellular level by cellular homeostasis kept with a contribution of both genetics and epigenetics. While genetics stands for the pool of biological information available to a cell, epigenetics can be translated as “above genetics”. This term encompasses all biochemical processes that influence gene expression without altering the DNA sequence. These processes activate or repress genes through modifications that do not change the genetic code but alter its structure; thus influencing how genes are translated and properly expressed in a cell. There is also a tight connection between genetic variants and epigenetics, being that the former influences the latter by impacting methylation and acetylation status at the DNA and histone levels.

It is estimated that around 20% of the variability in lifespan is attributed to genetic factors [2], while the other 80% stems from interactions among environmental and socio-economic influences that can affect the ageing process [3]. This is true if we do not refer to the hereditability of the exceptional longevity trait. Indeed, the genetic component is raised if we consider higher ages, as the probability of male siblings of centenarians was at least 17 times more likely to reach the age of 100 themselves as compared to the general population [4].

In this review, we will talk about achieving a health span through a full understanding of the both genetic and epigenetic changes underlying ageing and ageing-related diseases (Figure 1).

## 2. The Genetics of Longevity

The genetics of longevity is a rapidly developing field that attempts to answer the ancient question about the secret of human longevity. The first milestone was to establish to what extent longevity was inherited and so genetically determined as discussed by Perls et al. [4]. Thus, if genetics is driving human exceptional longevity, which genes and how, and eventually, which of their variants, contribute to cellular and organism homeostasis? It is important to dwell on this point, considering that, according to De Benedictis and Franceschi, “longevity” refers to the average lifespan under ideal conditions, most likely due to genetic traits [5]. Are these genes involved mainly in stress resistance, immune response, and metabolic regulation? Genetic association studies (mostly genome-wide association studies) have revealed that centenarians hold a distinct genetic profile compared to the general population with a notable enrichment in protective alleles of single-nucleotide polymorphisms (SNPs) [6].

Here, we will briefly examine the most important and well-known genes associated with longevity (i.e., Forkhead Box O3 (*FOXO3*), Sirtuin 1 (*SIRT1*), Apolipoprotein E (*APOE*), and Growth Hormone 3 (*GH3*)), with a focus on a recently discovered Longevity-Associated Variant-BPI Fold Containing Family B Member 4 (*LAV-BPIFB4*).”

The selection of these longevity-associated genes has been carried out through the literature on the assumption that only these genes showed a strict correlation between the genetic variants and the related pathways linked to the longevity trait, as well reviewed by Smulders and Deelen [7]. Indeed, *FOXO3*, *SIRT1*, and *GHR* are mainly involved in the insulin/insulin-like growth factor 1 (IGF-1) signalling [8], *FOXO3* in the mTOR signalling [9], *APOE* and *BPIFB4* in immune function-associated signalling [10,11,12,13,14,15,16], and all pathways have been found to be remarkable for lifespan regulation. Of course, over the years, other candidate genes associated with longevity have been found. Nevertheless, related studies have not been replicated in independent cohorts or population-based studies, and no in-depth studies have been conducted on the specific variants of these genes [7]. Moreover, on the other hand, when variants were found, the molecular mechanisms of these candidate genetic variants have not been deepened through in vitro or in vivo models. For all these reasons, it seemed worthwhile to focus our attention on genetic (rare) variants for whom the functional consequences and clinical effects are well known to be implicated in human longevity.

One of the most extensively studied genes related to longevity is the FOXO3 gene. This gene encodes a transcription factor that regulates several biological processes, including inflammation, aerobic glycolysis, cardiometabolic and oxidative stress, apoptosis, and DNA repair. These functions help protect against ageing-associated diseases (i.e., cardiovascular diseases, type 2 diabetes, neurodegenerative disorders, and cancer) [17]. Several studies have shown that specific variants of *FOXO3*, mainly the G allele of the rs2802292 polymorphism, are associated with increased longevity in several populations, including American, Asian, and European independent cohorts [18]

Besides *FOXO3*, *sirtuins (SIRTs)* are crucial histone deacetylases involved in critical signalling pathways, and, mainly, *SIRT1* participates in regulating energy metabolism, inflammation, DNA repair, and cellular senescence [19]. The polymorphism rs7895833 appears to be over-represented in healthy older people [20]. Interestingly, *FOXO3* is a *SIRT1* target, and, in oxidative stress conditions, this interplay prevents apoptosis and leads to stress resistance, thus counteracting the pro-ageing effects of reactive oxygen species (ROS) [21].

Another candidate gene for longevity was identified by Schächter et al. [22] in the ApoE gene, which encodes a protein involved in lipid metabolism with major expression in the liver and brain, making it capable of regulating neurodegenerative and cardiovascular diseases [23]. The *ApoE* gene shows three different isoforms, namely *ApoE ε2*, *ApoE ε3*, and *ApoE ε4*, encoded by the ε2, ε3, and ε4 alleles, respectively, each having a distinct influence on health and longevity. Indeed, if the ε4 allele of ApoE-promoting atherosclerosis is less frequent in long-living individuals, the ε2 one is associated with increased lifespan considering its higher frequency in centenarians, where it is able to lessen the incidence of cardiovascular and neurodegenerative diseases [23].

Several findings highlighted the importance of the *GHR* gene and the Insulin-like Growth Factor 1 (IGF-1)-associated signalling in affecting longevity. In detail, the *d3-GHR* deletion genotype variant is a relevant genetic polymorphism that has a positive impact on human longevity [24]. The inhibition of the GH/IGF-1 axis has been linked to enhanced oxidative stress response mainly through less production of ROS. Thus, the *d3-GHR* deletion genotype showed that carriers of this variant live longer than the non-carriers. In addition, Donlon et al. showed that carriers of the single-nucleotide polymorphism rs4130113 of the *GHR* gene allow hypertension-affected carriers to live as long as normotensive subjects [25].

Lastly, long-living individuals are enriched of the homozygous genotype for the minor allele of rs2070325 polymorphic variant of the *BPIFB4* gene, which is in linkage disequilibrium with three other SNPs forming a haplotype named “the longevity-associated variant-LAV” [26]. *LAV-BPIFB4* showed potential to promote longevity and mitigate age-related disorders, being beneficial to cellular homeostasis [27]. Indeed, the LAV polymorphism is over-represented in centenarians, and, moreover, different cohorts of long-living individuals carrying an LAV genotype are less affected by chronic diseases and are more able to reach healthier ageing with respect to the average population. This is very likely due to the impact of *LAV-BPIFB4* in reducing inflammation [12], in shaping immune response [11,13,14], in protecting against cardiovascular diseases [28,29,30,31] and neuroinflammation [15,32], in reducing the senescence profile [33], and in rejuvenating the immune system and vasculature [16]. In contrast, another isoform of *BPIFB4* (i.e., the rare genetic variant RV-BPIFB4) has been found to be enriched in frail subjects and to be responsible for cardiac dysfunctions [34], emphasizing the importance of the LAV isoform.

To be noted, the LAV haplotype was identified by analyzing homozygous genotypes in three independent populations (Italian, German, and North American Caucasian) while no differences were identified at the allelic level. Is this an indication of exceptional longevity as a recessive trait? This would also explain why in genetic isolates, where the high level of consanguinity determines an increase in recessive traits, there is a high degree of centenarians [35].

To conclude, long-living individuals possess a protective genetic profile characterized by an enrichment of beneficial alleles, allowing them to be shielded from ageing-related disease and serve as a model for successful ageing, likely attributable to their unique genome.

## 3. Epigenetics and Ageing

Recent findings by David Sinclair’s group have challenged the traditional hypothesis on ageing, which attributed it mainly to the accumulation of mutations in the genome. New studies suggest instead that epigenetics plays a central role in this process.

In mice, experiments inducing low levels of damage to DNA double helices alter the epigenetic landscape, leading to cellular dysfunction and ageing. These results support the idea that the loss of epigenetic information is a key factor in mammalian ageing, suggesting that genetic and epigenetic damage influence each other, thereby accelerating cellular senescence [36].

Our understanding of human ageing has also received a further boost from the study of two rare genetic syndromes: Hutchinson–Gilford progeria syndrome and Werner syndrome [37,38]. These conditions are characterized by premature ageing and a markedly reduced lifespan and are caused by mutations in genes associated with DNA repair mechanisms or laminin A. These mutations lead to disorganized chromatin structures, further suggesting the inner link between ageing and notable epigenetic changes. These include modifications in DNA methylation (DNAm) and histone alterations, which are crucial for regulating gene expression and DNA damage [39]; this determines a progressive deterioration of the epigenomic architecture and scaffolding, resulting in substantial alterations in chromosomal and genomic integrity, as well as changes in gene expression patterns [40].

The main epigenetic changes associated with ageing include a global reduction in DNA methylation (hypomethylation), which is accompanied by hypermethylation at specific loci [41]. DNA methylation affects both coding regions and intergenic repetitive sequences (IRSs), such as the family of repetitive elements in primate genomes (Alu). Alu elements, which constitute approximately 13.7% of the human genome, are transposable elements containing numerous methylation sites (CpGs). About 25% of the genome’s methylation is bound to Alu, inhibiting their transposable activity. Decreased methylation of Alu is associated with age-related diseases such as osteoporosis, diabetes, and Alzheimer’s and is particularly significant between the ages of 34 and 68 [42,43]. This process of hypomethylation is also correlated with a reduction in the expression of DNMT1, an enzyme critical for maintaining DNA methylation patterns, the impairment of which contributes to cellular ageing.

In the juvenile stages of life, eukaryotic DNA is hypermethylated in repetitive elements, intergenic regions, and gene bodies, promoting genomic stability and limiting transcription in these areas. With ageing, increased hypomethylation can lead to the expression of genes that should normally remain silenced. The loss or acquisition of methylation can alter gene expression, contributing to diseases and types of cancer.

Ageing is also associated with chromatin de-condensation, which alters the structure of heterochromatin and promotes the accumulation of errors. During the DNA damage repair process, the relocation of chromatin-modifying factors such as *SIRT1*, *SIRT6*, *HDAC1*, and *PARP1* occurs. It is believed that enzymes such as TET (Ten-Eleven Translocation) and DNMT (DNA Methyltransferase) are also delocalized in response to damage [36].

The over-proliferation of heterochromatin is driven by the formation of senescence-associated heterochromatin foci (SAHF); these lead to the silencing of genes promoting cell division. This process starts with the de-condensation of heterochromatin in the ageing organism, followed by the conversion of these heterochromatin regions into euchromatin regions [40]. Additionally, the loss of heterochromatin is accompanied by a decrease in H3K9me3 and the delocalization of heterochromatin protein 1 (HP1). These modifications, which typically promote the tighter binding of DNA to the histone complex to form heterochromatin, are gradually diminished during the ageing process [44]. During ageing, the loss of heterochromatin also leads to the activation of previously silent retrotransposons. These elements then transpose by reverse transcribing their transcripts into genomic cDNA copies that integrate into different locations in the genome, thus disrupting cellular homeostasis during ageing [45].

Accordingly, one of the key epigenetic phenomena in ageing is the heterochromatin loss model. This model describes the disruptions of heterochromatin domains, which are established during embryogenesis over the course of ageing, leading to alterations in global nuclear architecture, the de-repression of previously silenced genes, and aberrant patterns of gene expression [46].

Overall, typically, longevity is linked to global hypomethylation and local hypermethylation [47]. Interestingly, as discussed in the next sections, these epigenetic changes that help predict ageing might be the results of genetic variants and regulatory action of non-coding RNAs [48].

## 4. Genetic Variants Influencing DNA Epigenetic Modifications

Genetic variation also displays regulatory action on chromatin state, which influences gene expression, molecular interactions, and phenotypical traits. These can be both in *cis* (same chromosome) and in *trans* (across chromosomes) [49]. As previously discussed, DNAm is one of the most widely studied epigenetic modifications that capture the cumulative effects of environmental and genetic factors. Single-nucleotide polymorphisms (SNPs) associated with DNAm levels are known as methylation quantitative trait loci (meQTLs) [50], thus representing the best tool to quantify the effects of genetic variants on DNA methylation levels. Recent observations revealed that compared with SNPs that are not meQTLs (non-meQTLs), meQTLs were enriched in important regulatory regions, such as active promoters and strong enhancers; conversely, they were absent in regions with few active genes, including intergenic regions and regions with heterochromatic characteristics [51]. The genetic influences on DNA methylation were investigated at five different life stages in human blood: children at birth, childhood, and adolescence and their mothers during pregnancy and middle age [51]; so, actually, no clear correlation data are available for human ageing. However, meQTL data from mouse strains belonging to the BXD family (intercross mice derived from a cross between two strains, C57BL/6J (a C57 strain) and DBA/2J (a D strain)) revealed a regulatory hotspot on chromosome 5 that had the highest density of tra [52]. Specifically, genetic variation in the trans-meQTL hotspot, meQTL.5a (Hepatocyte Nuclear Factor 1 Alpha (HNF1A) locus), was able to influence CpG methylation at multiple developmental genes (e.g., Tet3, Jarid2, Sox2, Hnf4a, Kdm6b, Prdm16, and Foxa2). The pleiotropic influence of the D allele increases methylation at sites with typically low methylation during youth, while the B allele is associated with increased body weight and lipid levels. The methylation of the target CpGs, which are also under the convergent influence of ageing and diet, then contributes to variation in the survival trait, with the *D* allele associated with a slightly shorter lifespan.

In conclusion, the *HNF1A* locus is an example of how the variation in methylation could contribute to genotype-dependent variation in lifespan. Consistent with this, targets of HNF1A include the metabolic and longevity gene, *Igf1* (insulin-like growth factor 1). A GWAS study also found that a variant in *HNF1A* (rs6489785) is one of the 169 variants that jointly contribute to human longevity.

Likewise, through a genome-wide association carried out in Europeans and South Asians separately, Hawe Js et al. joined forces to search for other sequence variations disturbing DNA methylation. Among 11.2 million unique SNP–CpG associations in peripheral blood operating in *trans* and that comprise pairwise relationships between 1847 genetic loci and 3020 methylation loci, they highlighted the SNP rs174548, which is mapped on *FADS1*, a key enzyme in the metabolism of fatty acids that has a profound effect on methylation at cg21709803. Further, the strongest association was achieved with methylation in CD8^+^ T cells, which may help explain the relationship of this locus with inflammatory diseases [53]. Through the same approach, the genetic variation at the *NFKBIE* locus has been linked to rheumatoid arthritis through the *trans*-acting regulation of DNA methylation by nuclear factor kappa-light-chain-enhancer of activated B cells (NF-κB) [53].

Previous studies have reported the existence of a substantial proportion of meQTLs that exhibit a shared pattern across diverse cell types. Now, we are aware of the importance of investigating cell-type-specific methylation regulation. In this context, through a hierarchical Bayesian interaction (HBI) model, the highest number of significant meQTLs was found in CD4+ T cells, CD14+ Monocytes, B cells, and Natural killer cells [50], which may be dysfunctional during immune ageing. Accordingly, the variant rs2395178 in the *HLA-DRA* gene was identified as a CD8+ T-cell-specific meQTL for cg00886432 (*p*  =  5.46 × 10^−12^). As expected, we observed that rs2395178 showed a stronger correlation with DNAm in participants with a high abundance of CD8+ T-cells compared to a naïve response, a typical immune trait occurring in the elderly.

Concerning the *BPIFB4* gene in which homozygosity for a four single-nucleotide polymorphism (SNP) haplotypeencoding the Longevity-Associated Variant (LAV)—correlates with prolonged health span and reduced risk of cardiovascular (CV) complications and inflammation, *LAV-BPIFB4* itself can target epigenetic histone modification, leading to chromatin remodelling.

First of all, our observation that *BPIFB4* is more abundant in long-lived individuals (LLIs) as compared to young controls [27] was then supported by DNA methylomes in the crossroad between the newborn and the nonagenarian/centenarian groups developed by Heyn et al. [54]. By generating methylation profiles of 485,577 highly informative CpG sites, they finally showed a list of hypomethylated CpGs (*N*  =  1920), which included cg04087207 in the first exon of BPIFB4, indicating its possible overexpression.

Furthermore, the *LAV-BPIFB4* gene has been described to enhance the effect of butyrate on the acetylation of H3K9 while *WT-BPIFB4* was ineffective. To note, the transfection reagent and empty vector had no effect on the baseline expression level of H3K9ac compared to the untransfected HEK293 cells. Taken together, the data indicate a feedback loop between *LAV-BPIFB4* and butyrate [12].

Furthermore, in a cellular model of Huntington’s disease (HD), *LAV-BPIFB4* infection reduces nucleolar stress and DNA damage through the inhibition of the levels of H3K9me3, which are usually associated with transcriptional repression and heterochromatinization occurring in mature HD neurons, by accelerating histone clearance via the ubiquitin–proteasome pathway [32]. Finally, we also show that the systemic gene transfer of *LAV-BPIFB4* in aged C57BL/6J mice was associated with a significant reduction in the epigenetic clock-based biological age, as measured by a three-CpG clock method, showing SNP-CpG associations in multiple cell types [55].

All these examples clearly demonstrate that human ageing cannot be fully understood in terms of the constrained genetic setting; rather, a sort of epigenetic drift must be an alternative means of explaining age-associated alterations.

## 5. Modulation of Ageing by Non-Coding RNAs

As an additional mechanism, the ageing process also leads to alterations in non-coding RNA levels, which in turn modify the expression levels of age-related genes and increase age-related genomic instability, among them being microRNAs and long non-coding RNAs (LncRNAs) [56].

Certain ncRNAs that influence ageing include miR-28e3p and miR-126, which are found to be reduced in type 2 diabetes mellitus, as well as miR-146a and miR-21, both associated with the NF-κB pro-inflammatory pathway. The reduced expression of miR-146a has been linked to ageing, while miR-21 alterations are observed in age-related conditions such as cardiovascular diseases and osteoporosis [57].

MicroRNAs are small non-coding, single-stranded RNAs that can regulate gene expression by binding to the 3′ untranslated region (3′ UTR) of the target genes, and in doing so, there is the inhibition of the translation or the degradation of mRNA. MiRNAs may also impact other epigenetic events by regulating their associated enzymes. For instance, members of the miR-29 family modulate the expression of DNA methyltransferases (DNMTs) and Ten-Eleven Translocation (TET) enzymes in both healthy and pathological conditions [56]. LncRNAs are a diverse class of ncRNAs longer than 200 nucleotides. They can bind to DNA, RNA, or proteins and regulate gene expression mainly by post-transcriptional and post-translational modifications and chromatin remodelling [57]. Both ncRNAs have been linked to pathways associated with senescence and inflammation, including the p23/p21 pathway and the nuclear factor kappa-light-chain-enhancer of activated B cells (NF-κB). They have also been implicated in the development of neurological disorders, such as Huntington’s and Alzheimer’s diseases, as well as other age-related conditions like fibrosis, cardiovascular disease, and osteoporosis.

Additionally, nc886, also known as pre-miR-886, influences senescence by reducing the levels of p16INK4A and the cyclin-dependent kinase inhibitor p21 (p21Waf1/Cip1), both of which are common biomarkers of senescence, and by lowering ROS in fibroblast cell models. Based on this evidence, these molecules may serve as potential targets for anti-ageing therapies [57].

## 6. Inflammageing at the Interface Between Genetics and Epigenetics, a New Prognostic Factor for Human Longevity

The favourable skewing of immune response characterizing long-living individuals [58] emphasizes the importance of a balanced immune system throughout life. The successful genetic program of long-lived individuals modifies the course of the ageing process by enhancing an effective immune response, which can mitigate the harmful effects of inflammation and cardiovascular complications [12]. Among its many definitions, ageing can be defined as a process of imbalance, characterized by an increase and accumulation of random damage and by mechanisms, increasingly less efficient in their repair. Over time, this process leads to the development of chronic diseases, psychophysical frailty, and, eventually, death [59]. Among the various mechanisms of protection, the immune system plays a primary role in maintaining tissue homeostasis by eliminating cells that escape internal regulatory controls (e.g., cancer cells) or clearing the circulatory stream of factors that may cause tissue damage [60,61,62,63] activity and the pro-release of cytokines inflammatory [64,65,66] KDM6B (JMJD3), which is induced by inflammatory mediators such as IL-4 and TGF-β and regulates inflammatory and immune responses by removing the repressive epigenetic signs, thus triggering cellular senescence [67,68]. Integrative epigenomic association (EWAS) studies, which employ bioinformatic analysis of large datasets, have identified several epigenetic markers associated with indicators of circulatory inflammation. In particular, the relationship between mDNA and cytokines, both pro-inflammatory and anti-inflammatory, was examined in cohorts of elderly subjects [69]. One of the first studies focused on a cohort of 966 hypertensive African American participants, examining the association between circulating levels of C-reactive protein (CRP) and mDNA [70]. Another EWAS study was conducted by Levine et al., which aimed to assess the relationship between epigenetic ageing and inflammation. The study identified a positive association where high levels of CRP were associated with the increased transcription of inflammatory genes, including those involved in pathways associated with tumour necrosis factor-alpha (TNF-α) [71,72]. In a separate study involving two cohorts, one of European descent and the other African American, some CpG sites showed consistent associations with CRP in both cohorts. The pathway enrichment analysis of this methylamine signature indicated involvement in IL-6 and IL-9 signalling pathways. Several identified CpG sites have been mapped on inflammation-regulating genes, such as absent in melanoma 2 (AIM2)—which facilitates the assembly of the inflammasome by activating the caspase-1 and splitting pro-IL1 and pro-IL18—and suppressor of cytokine signalling 3 (SOCS3), a gene induced by cytokines such as IL-6, IL-10, and Interferon-gamma (IFNγ) [68,73].

## 7. The Lifestyle Factors for Successful Epigenetic Changes

In individuals who adopt a healthy lifestyle, characterized by balanced nutrition, caloric restriction, and regular physical exercise, the occurrence of age-related epigenetic events is less common, as this lifestyle promotes sustained health. Conversely, factors such as low levels of literacy, stress, a complex social network, poor financial resources, lack of sleep, the abuse of coffee and tea, and smoking have been associated with epigenetic modifications that accelerate the ageing process [42].

In this context, it is crucial to identify biomarkers and biological clocks that can highlight effective strategies in preventing and overcoming conditions that lead to poor health, especially since it has been shown that it is more cost-effective to increase the lifespan of individuals than to invest in disease-specific adaptations in older people [74].

Among the various lifestyle factors that influence healthy ageing, diet is one of the most relevant. Although different diets can promote healthy ageing, few have been shown to positively affect age-related conditions, such as the Mediterranean and the Okinawa diets, both associated with low levels of inflammation and oxidative stress, as well as a lower incidence of cancer and cardiovascular disease [75], inducing a reduction in epigenetic age based on the DNAGrimAge (DNA methylation-based biomarker) epigenetic clock, as reported in several epigenetic studies.

Furthermore, several compounds have shown the ability to induce epigenetic alterations, highlighting the role of nutrition in healthy ageing. In addition to compounds well known for their beneficial properties, such as lycopene in tomatoes, omega-3 fatty acids in fish oil and nuts, and resveratrol in wine and berries, other compounds have been described as inducing epigenetic changes. For example, betalain, present in red beetroot and characterized by antioxidant and anti-inflammatory activity, is able to increase the expression of genes involved in the DNAm demethylation of promoters of oncosuppressor genes; curcumin, a compound with anti-inflammatory effects and present in turmeric, can impact epigenetics modulating DNA methylation and the expression of tumour suppressor miRNA and regulating histone modifications through histone acetyltransferase (HAT) and histone deacetylase (HDAC); finally, hydroxytyrosol and oleic acid, found in olive oil and characterized by antioxidant functions, reduce low-density lipoprotein (LDL) cholesterol and modulate miRNA by increasing the expression of oncosuppressor and fatty acid biosynthesis miRNA and age-associated signalling.

Regarding caloric restriction, the reduction in caloric intake from 10 to 40% without compromising nutritional value has been associated with improvements in health and longevity [76]. Benefits include increased DNA repair and delayed age-related DNAm alterations, such as increased DNA Methyltransferase 3 Alpha (DNMT3a) immunoreactivity and reduced histone modifications through Sirtuin activation. It also slows down the neurodegeneration process in the Central Nervous System (CNS), improves glucose metabolism, and alters miR-125 activity involved in fat metabolism and longevity, resulting in a reduced incidence of diabetes, cancer, and other ageing-related and epigenetic diseases [77,78,79].

Although the cellular and molecular mechanisms are incompletely understood [80], regular physical exercise also has a positive impact on health, as it reduces frailty and improves cognitive abilities while delaying the progression of age-related DNA methylation alterations and inducing changes in miRNAs involved in the regulation of inflammation [81,82].

## 8. The Power of the Epigenetic Clocks as Prognostic and Diagnostic Biomarker of Ageing and Longevity

Studies have shown that epigenetic alterations can modulate the ageing process and alleviate its effects; they can delay the onset of age-related disorders and either increase or decrease lifespan. That is why understanding epigenetic modifications allows us to better understand the development of age-related diseases [83]. In this context, it is evident that epigenetic biomarkers play a significant role in the prognosis and diagnosis of ageing[41]. DNAm is the main epigenetic mechanism involved in ageing, characterized by genome-wide hypomethylation, hypermethylation of specific loci, and interindividual changes in DNAm values [84]. Studies have demonstrated that DNAm decreases with age in certain tissues in humans, mice, and cell cultures. Mice present a 5mC level reduction in various tissues through ageing, like the brain, liver, and small intestinal mucosa. The decrease in/loss of 5mC alters the physiological cell functions in older mice [85]. This implies that DNAm could be used as a biomarker to evaluate the “biological age” that does not always match with the chronological age that represents the number of years an individual has lived [86]. While chronological age can be used in forensic settings, biological age is used for monitoring the progression of a disease affecting the individual or the effect of a medical treatment [87], and for this reason, biological age is often referred to as the “epigenetic clock”, measuring changes in DNAm levels throughout the whole human genome [88].

The methylation status of the 28 million CpG dinucleotides present in the human genome changes with age [86], and even though the total amount of 5mC in the genome does not change with age, methylation at specific sites occurs [89]. Epigenetic clocks have allowed for the proper assessment of the efficacy of ageing interventions and in vitro have demonstrated that it is possible to reverse the epigenetic clock and to mitigate the phenotypic effects on ageing [88]. It is important, moreover, to understand the molecular mechanisms underlying ageing to identify potential targets that allow us to prevent multimorbidity and disability associated with age [41].

Bocklandt described the first epigenetic clock using the DNA extracted from human saliva and by performing the regression of the chronological age on the DNAm levels using a penalized regression model [90]. Also, Hannum et al. developed an age estimator based on 71 CpGs found on blood DNA, but in both cases, the epigenetic clocks presented limitations. Blood, in fact, undergoes changes in its composition that are age-related, and one of the limitations is that the epigenetic clock was created considering adult samples, possibly causing biases in the estimation of child samples or in the case of non-blood samples [41]. Moreover, it has been shown that the association between the epigenetic clock and mortality weakens as the accuracy of chronological age prediction increases; therefore, the necessity of assessing both chronological age and its association with morbidity was highlighted [91]. This led to the creation of second-generation clocks—PhenoAge and GrimAge—specifically designed to overcome these limitations. PhenoAge was developed using a two-stage approach in which the first one includes ten clinical characteristics: chronological age, albumin, creatinine, glucose, C-reactive protein, lymphocyte percentage, mean cell volume, red blood cell distribution width, alkaline phosphatase, and white blood cell count. These features were profiled to develop a phenotypic age estimator; then, in the second stage, this estimator was regressed on DNAm levels and this allowed the identification of 513 CpG sites significantly different in disease and mortality among individuals with the same chronological age [92,93].

Furthermore, a third-generation clock was created: the DunedinPoAm clock. This clock evaluates changes in 18 biomarkers related to organ system integrity among individuals that presented the same chronological age and by doing so, it measures the rate of ageing. It indicates how quickly ageing occurs in the years leading up to the measurement [94]. It has also been reported that CpG sites vary with age, and the first evidence showed that in monozygotic twins, the older one presents higher methylation variation in overall 5mC content [85].

Despite all the improvements in epigenetic clocks, they still have some important limitations; in fact, many epigenetic clocks are related to adult samples, under-representing subjects below the adult age, and, in addition to this, epigenetic clocks lose effectiveness in older age groups because the epigenetic drift increases, making the predictions really hard to be accurate [87].

Another useful tool is represented by the age acceleration (AA) index, which is the result of the difference between biological and chronological age, and so it represents the “ticking speed” of the clock. It is calculated by finding residuals of predicted epigenetic age regression on chronological age and indicates the rate at which ageing is occurring. Tissue-specific AA is associated with several age-related disorders such as cognitive dysfunction, cardiovascular disease, and frailty. Moreover, there are many factors that can cause differences in AA: sex chromosomes, hormones, and lifestyle. Smoking, for example, is a lifestyle factor that can modify DNA methylation patterns, increase the age acceleration in respiratory tissues, and is associated with frailty in older adults. Frail individuals have been shown to have lower levels of global methylation in peripheral blood when compared to prefragile and nonfragile individuals. In addition, those who were prefragile at baseline and became frail after 7 years of follow-up showed a significant reduction in DNA methylation compared to frail individuals at baseline. This indicates that the reduced methylation in frail individuals could lead to decreased gene silencing and to the activation of genes that were previously repressed, and so there is an alteration of gene expression. This was further confirmed by the increase in extrinsic epigenetic AA, which measures AA based on DNA methylation and has been associated with an increased risk of physical frailty [95].

## 9. Conclusions

Globally, life expectancy is 76 years for women and 71 years for men, with the highest levels observed in Australia, France, Italy, Japan, South Korea, Spain, Sweden, and Switzerland [96]. Furthermore, in 1990, there were only 95,000 people over 100 years old in the world; today, there are over half a million centenarians, and this number continues to rise.

In this brief review, we discussed genetic (Table 1) and epigenetic mechanisms (Table 2) as key contributors to the increasing elderly population, as well as the molecular determinants that can slow ageing and extend health span. Indeed, while environmental and lifestyle factors hold significance during youth, genetics seem to play a larger role in attaining extreme old age to increase lifespan. Epigenetic alterations associated with ageing significantly influence cellular functionality and resilience to stress. This examination underscores the critical importance of incorporating both genetic and epigenetic considerations into anti-ageing research.

Similarly, we now recognize the necessity of utilizing epigenetic biomarkers and molecular genetic tools to predict and manage various ageing-related conditions, particularly as the human ageing process is currently understood to be complex and nonlinear. It is widely recognized that the occurrence of ageing-related illnesses and the risk of death do not rise gradually but accelerate after specific time points, with musculoskeletal problems, Alzheimer’s, and cardiovascular disease risk exhibiting a sharp rise post-60. Recent studies have shown persistent nonlinear trends in molecular ageing markers, indicating significant dysregulation at two key points around 44 years and 60 years of chronological age [97]. This phenomenon can be attributed to nonlinear rates of RNA and protein expression, as well as nonlinear alterations in methylation status during ageing. Certainly, certain changes may be associated with lifestyle or behavioural influences; thus, the fine modulation of the power-law pattern of epigenetic events will play a crucial role in determining the quality of life for elderly individuals in the near future.

## Figures and Tables

**Figure 1 biology-14-00092-f001:**
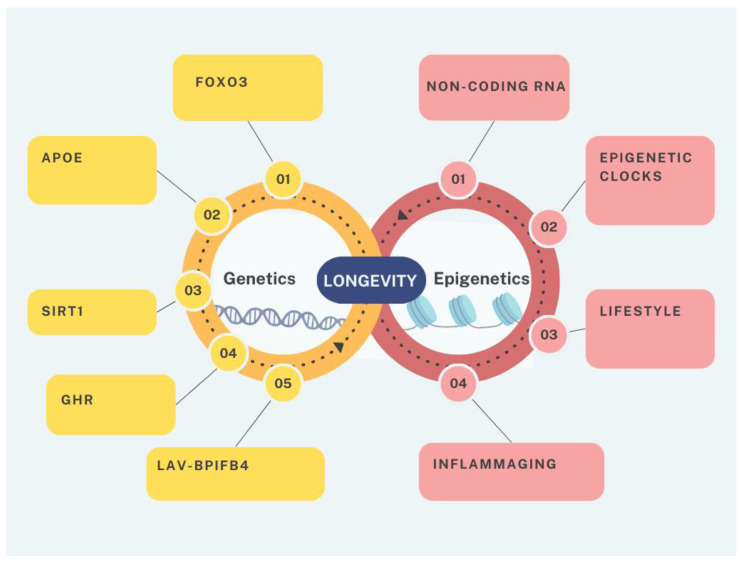
The genetic and epigenetic influences on human longevity.

**Table 1 biology-14-00092-t001:** Main genetic influences on human longevity.

Gene	Pathway	Variant/SNPs	Clinical Effects of the Variant	Experimental Model/Methods	Ref.
**FOXO3**	Inflammation, aerobic glycolysis, cardiometabolic and oxidative stress, apoptosis, DNA repair	*rs2802292*	Lower incidence of ageing-associated diseases (cardiovascular, neurodegenerative, and cancer)	GWAS in different cohorts	[18]
**SIRT1**	Energy metabolism, inflammation, DNA repair and senescence	*rs7895833*	Interplay with FOXO3 prevents from oxidative stress conditions and apoptosis	GWAS in 1 cohort	[20]
**APOE**	Lipid metabolism	*ε4*	Atherosclerosis	GWAS in different cohorts	[23]
		*ε2*	Lower incidence of ageing-associated diseases (cardiovascular and neurodegenerative)	GWAS in different cohorts	[23]
**GHR**	Interplay with IGF-1 for oxidative stress response	*d3-GHR deletion*	Enhanced oxidative stress response (less ROS production)	GWAS in different cohorts	[24]
**BPIFB4**	Immune function, inflammation, cellular homeostasis,	*rs2070325* *(LAV-BPIFB4)*	Reducing inflammation and neuroinflammation, shaping and rejuvenating immune system, reducing the senescence profile	GWAS in different cohorts; in vivo and in vitro models	[11,13,14,15,16,26,27,28,29,30,31,32,33]
		*rs2070325* *(RV-BPIFB4)*	Enhancer of frailty and cardiac dysfunctions	GWAS in different cohorts; in vivo and in vitro models	[34]

**Table 2 biology-14-00092-t002:** Main epigenetics changes influencing human longevity.

Epigenetic Change	Mechanism	Clinical Effects	Experimental Model/Methods	Ref.
**DNA hypomethylation**	Decreased Methylation of Alu	Osteoporosis, diabetes, Alzheimer’s disease	Analytical cross-sectional and in vitro studies	[42,43]
	Reduction of DNTM1	Reduced aging-associated transcriptional changes; interneuron survival and slight autoimmunity	In vivo C57BL/6 aged murine model	[40,41]
**Heterocromatin de-condensation**	Loss of the H3K9me3	Degeneration of multiple organs leading to premature aging	In vivo TKOCAGCre murine model	[42]
	Activation of silent retrotransposons	Premature aging and neurodegeneration	In vivo and ex vivo C57BL/6 aged murine models	[43,44]
**Non-coding RNAs**	Deletion of miR-29 (modulation of DNMT3A and TET enzymes)	Repression of genes associated with neuronal activity	In vivo murine models	[53]
	nc886 (Reduction of p16^INK4A^, p21 and ROS)	Decreased senescence	In vitro fibroblast cell models	[54]

## Data Availability

Data, materials, and protocols are available upon request by email to the corresponding authors due to privacy/ethical restrictions.

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
