# Peer review of "The Genetic and Epigenetic Arms of Human Ageing and Longevity"

_biology, 2025, doi:10.3390/biology14010092_

Round 1

Reviewer 1 Report

Comments and Suggestions for Authors

The review addresses a hot topic in the areas of ageing by investigating studies that have some information on the major genetic and epigenetic contributions to the ageing process and longevity of individuals.

Major comment

The author should prepare a table that will summary the key findings from the review of studies. For example, a list of genetic variants and their impacts, the specific epigenetic markers associated with ageing, the methods used in the studies to derived the information/association. Then, the relevance to the inflammageing will make sense. Otherwise, it will be challenging for most readers to learn and derive vital information from this work.      

Author Response

Sure, as judged very informative, we decided to insert 2 additional tables, one summarizing the genes, pathways and clinical effects of gene variants associated to the longevity and the other one that includes mechanism of epigenetic changes and related clinical effects linked to the longevity. In both, experimental model and methods used are also indicated.

Reviewer 2 Report

Comments and Suggestions for Authors

The authors of the review have done a great job. The review is certainly interesting. But there are aspects that need to be supplemented and clarified. It is unclear on what principle the authors selected publications for inclusion in the review. Not enough attention is paid to such factors as telomere length and mitochondrial DNA copy number. The authors highlighted only a few genes. Why exactly these? That is, it is necessary to explain the criteria by which these genes were selected and others were excluded.

Author Response

We thank the reviewer for his/her comment which allows us to clarify a fundamental point that we will include in the manuscript.

The selection of these longevity-associated genes has been carried out through literature on the assumption that only these genes showed a strict correlation between the genetic variants and the related pathways linked to the longevity trait, as well-reviewed by Smulders and Deelen (doi: 10.1111/joim.13740). Indeed, FOXO3A, SIRT1, and GHR are mainly involved in the insulin/insulin-like growth factor 1 (IGF-1) signaling  (doi.org/10.1007/s11357-011-9340-3), FOXO3A in the mTOR signaling (doi.org/10.1111/acel.12015), APOE and BPIFB4 in immune function-associated signaling [doi:10.1155/2010/186813, ref in the manuscript 18- 21, 26, 29], all pathways found to be remarkable for lifespan regulation. Of course, over the years, other candidate genes associated with longevity have been found. Nevertheless, related studies have not been replicated in independent cohorts or population-based studies and no in-depth studies have been conducted on the specific variants of these genes (doi: 10.1111/joim.13740). Moreover, on the other hand, when variants were found, the molecular mechanisms of these candidate genetic variants have not been deepened through in vitro or in vivo models. For all these reasons, it seemed worthwhile to focus our attention on genetic (rare) variants for whom the functional consequences and clinical effects are well known to be implicated in human longevity.

For the same reasons, while there is a large body of correlative data linking both telomere length and mitochondrial DNA copy number to aging, a direct role in longevity is lacking and deserves further attention in a separate review.

Reviewer 3 Report

Comments and Suggestions for Authors

Article Summary:

The article, titled “The genetic and epigenetic arms of human ageing and longevity, explores the dynamic relationship between genetic and epigenetic mechanisms in the context of human ageing and longevity. It addresses critical topics, including genetic variants influencing epigenetic modifications, the modulation of ageing by non-coding RNAs, and the concept of inflammageing as a prognostic factor for human healthspan. The manuscript highlights the role of lifestyle factors in shaping successful epigenetic changes and discusses epigenetic clocks as powerful biomarkers for evaluating biological age and predicting ageing-related conditions. A particularly innovative aspect of the study is its examination of how nonlinear molecular patterns, such as RNA and protein expression and methylation changes, influence the ageing process at specific chronological milestones. The conclusion effectively ties together these insights, emphasising the necessity of integrating genetic and epigenetic considerations into anti-ageing research. While the article is comprehensive, certain technical terms could be clarified to improve accessibility.

Review:

The manuscript presents a well-rounded and timely exploration of the genetic and epigenetic mechanisms underlying human ageing and longevity, incorporating recent advances in molecular biology and biomarker research. Its discussion of epigenetic clocks and inflammageing is particularly innovative, addressing emerging trends in ageing research. Nonetheless, there are areas where the manuscript could be improved, such as language and clarity, as well as structural cohesion. Furthermore, a review by a professional editor or native English speaker is recommended to refine the text and enhance readability. Despite these areas for improvement, the article provides a meaningful contribution to our understanding of ageing and longevity, offering insights that could inform future research and clinical applications.

The manuscript titled “The genetic and epigenetic arms of human ageing and longevity” presents significant findings. However, to strengthen the data and ensure clarity, the suggested revisions should be fully addressed.

Major revisions

Gene names are often written in italics to distinguish them from other elements. For example, FOXO3 and SIRT1. Proteins encoded by genes, on the other hand, are not written in italics, e.g., FOXO3 and SIRT1. This rule was established to facilitate the differentiation between genes (genomic elements) and gene products (proteins). Therefore, I strongly recommend that authors format all gene abbreviations used in the manuscript in italics. This criterion must also be met for Figure 1.

Minor revisions

Line 33:

Rewrite the sentence for clarity:

“… contributing the recent extended lifespan.” to “… contributing to the recent increase in lifespan.”

Lines 33 to 36:

Rewrite the sentence for clarity:

“In addition, the right healthy-lifestyle approaches also contributes to delay ageing and increase longevity, thus explaining the increased life-span observed in the latest century in industrialized countries.” to “In addition, adopting healthy lifestyle approaches contributes to delaying ageing and increasing longevity, thus explaining the increased lifespan observed over the past century in industrialized countries.”

Lines 33 to 36:

Rewrite the sentence for clarity:

“Thus, a person's environment and life-style influence how we age together with genetics.” to “Thus, a person's environment and lifestyle, along with genetics, influence how they age.”

Lines 43 to 46:

Rewrite the sentence for clarity:

“This term encompasses all biochemical processes that influence gene expression without altering the DNA sequence, by activating or repressing them; these are modifications of DNA that do not change the genetic code but alter its structure and affect how genes are translated and properly expressed in a cell.” to “This term encompasses all biochemical processes that influence gene expression without altering the DNA sequence. These processes activate or repress genes through modifications that do not change the genetic code but alter its structure, thereby affecting how genes are translated and expressed in a cell.”

Lines 60 to 61:

Rewrite the sentence for clarity:

“Genetics of longevity is a quickly developing field that attempt to answer to ancient question on the secret of human longevity.” to “The genetics of longevity is a rapidly developing field that attempts to answer the ancient question about the secret of human longevity.”

Lines 68 to 71:

Rewrite the sentence for clarity:

“Genetic association studies (mostly Genome-wide association studies) revealed that centenarians hold a distinct genetic profile compared to the general population with a notable enrichment in protective alleles of single nucleotide polymorphisms (SNPs).” to “Genetic association studies, particularly genome-wide association studies (GWAS), reveal that centenarians possess a distinct genetic profile compared to the general population, characterized by a notable enrichment in protective alleles of single nucleotide polymorphisms (SNPs).”

Lines 72 to 74:

Rewrite the sentence for clarity:

“Here, we will briefly look into the most important and well-known genes associated with longevity (ie, FOXO3, SIRT1, APOE, GH3) with a focus on a recently discovered longevity-associated gene variant (ie, LAV-BPIFB4).” to “Here, we will briefly examine the most important and well-known genes associated with longevity (i.e., Forkhead Box O3 (FOXO3), Sirtuin 1 (SIRT1), Apolipoprotein E (APOE), and Growth Hormone 3 (GH3)), with a focus on a recently discovered Longevity-Associated Variant – BPI Fold Containing Family B Member 4 (LAV-BPIFB4).”

Line 73:

Shouldn’t the term “GH3”, which refers to a specific gene, be “GHR” (as mentioned in line 98)? If this is not a typographical error, please provide a conceptual difference between the two terms.

Lines 75 and 76:

Rewrite the sentence for:

“FOXO3 (i.e. forkhead box O3)” toFOXO3

Lines 76 to 80:

Rewrite the sentence for clarity:

“This gene encodes for a transcription factor that exerts regulatory effects on several biological processes such as inflammation, aerobic glycolysis, cardiometabolic and oxidative stress, apoptosis, and DNA repair, preserving from ageing-associated diseases (ie, cardiovascular diseases, type 2 diabetes, neurodegenerative disorders, and cancer) [7].” to “This gene encodes a transcription factor that regulates several biological processes, including inflammation, aerobic glycolysis, cardiometabolic and oxidative stress, apoptosis, and DNA repair. These functions help protect against aging-associated diseases, such as cardiovascular diseases, type 2 diabetes, neurodegenerative disorders, and cancer [7].”

Line 88:

Rewrite the sentence:

ROStoreactive oxygen species (ROS)”

Line 88:

Rewrite the sentence:

“In mice, experiments of induction of low degree of damage to DNA double helices, alters the epigenetic landscape, …” to “In mice, experiments inducing low levels of damage to DNA double helices alter the epigenetic landscape, …”

Line 146:

Rewrite the sentence:

“DNAm” to “DNA methylation (DNAm)”

Lines 152 and 153:

Rewrite the sentence:

“DNA methylation affects both coding regions and intergenic repetitive sequences (IRS) of the genome, such as Alu. Alu, which make up approximately 13.7% …” to “DNA methylation affects both coding regions and intergenic repetitive sequences (IRS), such as the family of repetitive elements in primate genomes (Alu). Alu elements, which constitute approximately 13.7% …”

Line 193:

Rewrite the sentence:

4. Genetic variants influencing DNA epigenetic modifications. At the end of the topic, delete (.).

Line 197:

Rewrite the sentence:

“DNA methylation (DNAm)” to “DNAm”

Line 203:

Rewrite the sentence:

“promotors” to “promoters”

Line 206:

Rewrite the sentence:

“was” to “were”

Lines 208 to 211:

Rewrite the sentence:

“However, MeQTLs data from mouse strains belonging to the BXD family revealed regulatory hotspot on chromosome 5 that had the highest density of trans-acting methylation QTLs (trans-meQTLs) associated with multiple distant CpGs in an age-dependent manner [48].” to “However, MeQTLs data from mouse strains belonging to the BXD family (intercross mice derived from a cross between two strains, C57BL/6J (a C57 strain) and DBA/2J (a D strain)) revealed regulatory hotspot on chromosome 5 that had the highest density of trans-acting methylation QTLs (trans-meQTLs) associated with multiple distant CpGs in an age-dependent manner [48].”

Line 212:

Rewrite the sentence:

“Hnf1a” to “Hepatocyte Nuclear Factor 1 Alpha (HNF1A)”

Lines 214 to 216:

Rewrite the sentence:

“The pleiotropic influence results in the D allele increasing methylation at sites that typically have low methylation when young, and the B allele increasing body weight and lipid levels.” to “The pleiotropic influence of the D allele increases methylation at sites with typically low methylation during youth, while the B allele is associated with increased body weight and lipid levels.”

Line 219:

Rewrite the sentence:

“HNFs” toHNF1A

Line 234:

Rewrite the sentence:

“NF-κB” to “nuclear factor kappa-light-chain-enhancer of activated B cells (NF-κB)”

Line 245:

Rewrite the sentence:

… gene in which, , delete (,).

Line 247:

Rewrite the sentence:

“CV” to “cardiovascular (CV)”

Line 250:

Rewrite the sentence:

“First of all our…” to “First of all our,…”

“LLIs” to “long-lived individuals (LLIs)”

Line 261:

Rewrite the sentence:

“mHTT” to “mutant Huntingtin (mHTT)”

Line 285:

Rewrite the sentence:

“3’ UTR” to “3’ untranslated region (3’ UTR)”

Line 288:

Rewrite the sentence:

 “DMNTs and TET” to “DNA methyltransferases (DNMTs) and Ten-Eleven Translocation enzymes (TET)”

Lines 299 and 300:

Rewrite the sentence:

 “reactive oxygen species (ROS)” to “ROS”

Line 305:

Rewrite the sentence:

 “…immune system throughout the life.” to “…immune system throughout life.”

Lines 310 and 311:

Rewrite the sentence:

 “…increasingly less efficient, devoted to their repair.” to “…progressively less efficient in their repair mechanisms.”

Lines 312 to 319:

Rewrite the sentence:

“Among the various mechanisms of protection, the immune system plays a primary role in monitoring that contributes to tissue homeostasis by eliminating cells that escape internal regulatory controls, e.g. cancer cells, or clean the circulatory stream of factors that may cause tissue damage [56-59] activity and pro-release of cytokines inflammatory[60-62] KDM6B (JMJD3), which is induced by inflammatory mediators such as IL-4 and TGF-β, regulates inflammatory and immune responses by removing the repressive epigenetic signs, thus triggering cellular senescence [63, 64].” to “Among the various mechanisms of protection, the immune system plays a primary role in maintaining tissue homeostasis by eliminating cells that escape internal regulatory controls (e.g., cancer cells) or clearing the circulatory stream of factors that may cause tissue damage [56-59]. The activity and pro-release of inflammatory cytokines [60-62], as well as KDM6B (JMJD3), which is induced by inflammatory mediators such as interleukin-4 (IL-4) and transforming growth factor beta (TGF-β), regulate inflammatory and immune responses by removing repressive epigenetic marks, thereby triggering cellular senescence [63, 64].”

Lines 319 and 320:

Rewrite the sentence:

“…bioinformatic analysis on many data,…” to “…bioinformatic analysis of large datasets,…"

Lines 327 to 329:

Rewrite the sentence:

“...in which high levels of CRP were correlated with increased transcription of inflammatory genes, including those involved in pathways associated with TNF-α [67, 68].” to “...where high levels of CRP were associated with increased transcription of inflammatory genes, including those involved in pathways associated with tumor necrosis factor-alpha (TNF-α) [67, 68].”

Line 331:

Rewrite the sentence:

“PCR” to “CRP”

Line 333:

Rewrite the sentence:

“AIM2” to “absent in melanoma 2 (AIM2)”

Line 335:

Rewrite the sentence:

“SOCS3” to “suppressor of cytokine signaling 3 (SOCS3)”

Lines 335 and 336:

Rewrite the sentence:

IFNγtoInterferon-gamma (IFNγ)

Line 340:

Rewrite the sentence:

“…since this lifestyle promotes the duration of health.” to “…as this lifestyle promotes sustained health.”

Lines 344 and 345:

Rewrite the sentence:

“…can high for preventing and overcoming those conditions…” to “…can help in preventing and overcoming conditions…”

Lines 350 and 351:

Rewrite the sentence:

“…Mediterranean diet and the Okinawa diet…” to “…Mediterranean and Okinawa diets…”

Line 353:

Rewrite the sentence:

“DNAGrimAge” to “DNAGrimAge (DNA methylation-based biomarker)”

Line 359:

Rewrite the sentence:

“…betalaina…” to “…betalain…”

Line 364:

Rewrite the sentence:

“HAT and HDAC” to “histone acetyltransferase (HAT) and histone deacetylase (HDAC)”

Lines 364 and 365:

Rewrite the sentence:

“…hydroxytyrosol oleic acid, found in olive oil and characterized by antioxidant functions,…” to “…hydroxytyrosol and oleic acid, found in olive oil and characterized by antioxidant properties,…”

Line 366:

Rewrite the sentence:

“LDL” to “low density lipoprotein (LDL)”

Line 371:

Rewrite the sentence:

“DNMT3a” to “DNA Methyltransferase 3 Alpha (DNMT3a)”

Line 373:

Rewrite the sentence:

“CNS” to “Central Nervous System (CNS)”

Lines 377 to 379:

Rewrite the sentence:

“…reducing frailty and improves cognitive abilities, delaying the progression of age-related DNA methylation alterations…” to “…reducing frailty and improving cognitive abilities while delaying the progression of age-related DNA methylation alterations…”

Lines 383 and 385:

Rewrite the sentence:

“Studies have shown that epigenetic alterations can modulate the ageing processes and alleviate it, they can delay the onset of age-related disorders and lead to an increase or a decrease of the lifespan.” to “Studies have shown that epigenetic alterations can modulate the ageing process and alleviate its effects, they can delay the onset of age-related disorders and either increase or decrease lifespan.”

Line 387:

Rewrite the sentence:

 “…evident how epigenetic biomarkers have a significant role for the prognosis and diagnosis...” to “...evident that epigenetic biomarkers play a significant role in the prognosis and diagnosis...”

Line 391:

Rewrite the sentence:

“…tissues both in humans and mice, but also in cell cultures." to “…tissues in humans, mice, and cell cultures.”

Lines 402 and 403:

Rewrite the sentence:

“…and even if the total amount of 5mC in the genome doesn’t change with age, so it occurs for the methylation at specific sites [83].” to “…and even though the total amount of 5mC in the genome doesn’t change with age, methylation at specific sites occurs [83].”

Lines 403 and 404:

Rewrite the sentence:

“Epigenetic clocks allowed to properly assess the efficacy of ageing interventions and in vitro studies…” to “Epigenetic clocks have allowed for the proper assessment of the efficacy of ageing interventions and in vitro studies…”

Line 409:

Rewrite the sentence:

In the sentence: “Bocklandt described the first epigenetic clock using the DNA extracted from saliva…” The authors need to specify the origin of the saliva: is it from humans, or another specimen?

Lines 417 to 419:

Rewrite the sentence:

“…therefore it was highlighted the necessity of assessing both chronological age and its association with morbidity [85].” to “…therefore, the necessity of assessing both chronological age and its association with morbidity was highlighted [85].”

Line 428:

Rewrite the sentence:

“third generation” to “a third-generation”

Lines 456 and 457:

Rewrite the sentence:

“…who’s been associated with increased risk of being physically frail [89].” to “…which has been associated with an increased risk of physical frailty [89].”

Line 461:

Rewrite the sentence:

“Life expectancy is globally 76 for women and 71 for men with the highest levels in…” to “Globally, life expectancy is 76 years for women and 71 years for men, with the highest levels observed in…”

Line 464:

Rewrite the sentence:

“…million, and rising.” to “…million centenarians, and this number continues to rise.”

Lines 465 to 467:

Rewrite the sentence:

“In this short review we discussed the genetic and epigenetic mechanisms as the main foundation of the rise in elderly population but also the molecular determinants to slow ageing and extend the healthspan.” to “In this brief review, we discussed genetic and epigenetic mechanisms as key contributors to the increasing elderly population, as well as the molecular determinants that can slow ageing and extend healthspan.”

Line 469:

Rewrite the sentence:

“…to increase the lifespan.” to “…to increase lifespan.”

Lines 469 and 470:

Rewrite the sentence:

“Further epigenetic alterations with ageing significantly influence cellular functionality and resilience to stress.” to “Epigenetic alterations associated with ageing significantly influence cellular functionality and resilience to stress.”

Lines 474 and 476:

Rewrite the sentence:

 “…particularly given that the human ageing process is currently understood as a complex, not linear trend.” to “…particularly as the human ageing process is currently understood to be complex and non-linear.”

Lines 481 and 483:

Rewrite the sentence:

“Readers can now comprehend that the foundation of this phenomenon is attributed to nonlinear rates of RNA and protein expression, along with nonlinear alterations in methylation status during ageing.” to “This phenomenon can be attributed to nonlinear rates of RNA and protein expression, as well as nonlinear alterations in methylation status during ageing.”

Lines 481 and 483:

Rewrite the sentence:

“…thus a fine modulation of the power law pattern of epigenetic events will determine the quality of life for elderly individuals shortly.” to “thus, fine modulation of the power-law pattern of epigenetic events will play a crucial role in determining the quality of life for elderly individuals in the near future.”

Author Response

   We thank the reviewers for his/her valuable comments. All the sentences and technical terms has been properly corrected throughout the text